# The Combining Ability and Heterosis Analysis of Sweet–Waxy Corn Hybrids for Yield-Related Traits and Carotenoids

**DOI:** 10.3390/plants13020296

**Published:** 2024-01-18

**Authors:** Kanyarat Prai-anun, Yaowapha Jirakiattikul, Khundej Suriharn, Bhornchai Harakotr

**Affiliations:** 1Department of Agricultural Technology, Faculty of Science and Technology, Thammasat University, Pathum Thani 12120, Thailand; kanyarat.prai@dome.tu.ac.th (K.P.-a.); yjirakia@tu.ac.th (Y.J.); 2Department of Agronomy, Faculty of Agriculture, Khon Kaen University, Khon Kaen 40002, Thailand; sphala@kku.ac.th

**Keywords:** biofortification, gene action, heritability, general and specific combining ability, hybrid vigor, hybrid breeding, *Zea mays* L.

## Abstract

Improving sweet–waxy corn hybrids enriched in carotenoids via a hybrid breeding approach may provide an alternative cash crop for growers and provide health benefits for consumers. This study estimates the combining ability and heterosis of sweet–waxy corn hybrids for yield-related traits and carotenoids. Eight super sweet corn and three waxy corn lines were crossed to generate 24 F_1_ hybrids according to the North Carolina Design II scheme, and these hybrids were evaluated across two seasons of 2021/22. The results showed that both additive and non-additive genetic effects were involved in expressing the traits, but the additive genetic effect was more predominant. Most observed traits exhibited moderate to high narrow-sense heritability. Three parental lines, namely the ILS_2_ and ILS_7_ females and the ILW_1_ male, showed the highest positive GCA effects on yield-related traits, making them desirable for developing high-yielding hybrids. Meanwhile, five parental lines, namely the ILS_3_, ILS_5_, and ILS_7_ females and the ILW_1_ and ILW_2_ males, were favorable general combiners for high carotenoids. A tested hybrid, ILS_2_ × ILW_1_, was a candidate biofortified sweet–waxy corn hybrid possessing high yields and carotenoids. Heterosis and per se performance were more positively correlated with GCA_sum_ than SCA, indicating that GCA_sum_ can predict heterosis for improving biofortified sweet–waxy corn hybrid enriched in carotenoids. The breeding strategies of biofortified sweet–waxy corn hybrids with high yield and carotenoid content are discussed.

## 1. Introduction

People in most Asian countries commonly consume waxy or glutinous corn (*Zea mays* L. var. *ceratina*). In Thailand, people harvest waxy corn during the immature stage and consume it as boiled or steamed corn, like sweet corn [1,2]. Traditional waxy corn has more significant amounts of amylopectin (95–100%) [3], resulting in high stickiness and soft tenderness but poor sweetness and low sugar content [4]. Corn breeders attempt to develop new waxy corn hybrids with high yields, unique eating qualities, and uniform ear appearance [5]. Sweet–waxy corn hybrids can improve the palatability of traditional cooked waxy corn by utilizing the synergistic effect of multiple sweet genes, including *su_1_*, *sh_2_*, and *se*, into the *wx* background [5,6,7]. Generally, waxy corn has various kernel colors, including white, white-cream, yellow, purple, and black, relating to nutraceutical compounds such as carotenoids, anthocyanins, and phenolics that promote human health [1,8,9,10]. However, commercial varieties with white or white-cream kernel colors, lacking carotenoids, are preferable in many countries, and consumers do not prefer other kernel colors [11,12]. The consumer acceptance of other waxy corn kernel colors, for instance, yellow, is challenging. In contrast, yellow sweet corn, the most popular corn type in the market, has been recognized as a good source of macular carotenoids, including lutein and zeaxanthin [13,14]. These two carotenoids, called macular pigments, which humans cannot synthesize but should accumulate from dietary foodstuffs, may improve vision and prevent age-related macular degeneration (AMD) and blue-light damage [15,16,17]. Considering that health aspect, the University of Queensland successfully improved a new super sweet corn hybrid to provide an adequate intake of zeaxanthin per cob per day, which is equivalent to synthetic supplements of 2 mg per day as suggested [15,18]. Other carotenoids found in biofortified corn are provitamin A, including α-carotene, β-carotene, and β-cryptoxanthin, which can be converted into retinol [19]. Those compounds have several essential health benefits, such as inhibiting some forms of cancer, preventing macular degeneration, decreasing the risk of cataract formation, preventing cardiovascular disease, and enhancing immunity [11,20]. Therefore, providing biofortified sweet–waxy corn hybrids with high eating quality and carotenoid contents will expand the market segments and benefit human health.

The use of heterosis breeding offers the possibility of improving the quantitative traits in cross-pollinated crops. Sunny et al. [21] reported that per se evaluation is often ineffective for parental selection on yield-related traits due to their polygenic nature. The selection can be more biased due to unstable performance across environments and weaker vigor due to inbreeding depression [22]. Understanding the effects of general combining ability (GCA) and specific combining ability (SCA) between inbred lines and optimizing heterosis in their hybrids for yield-related traits and carotenoids is critical to heterosis-based biofortification breeding [11,23]. Combining ability analysis can also assess the relative importance and modes of gene action involved in the commercial hybrids to desired traits [24]. While our previous study demonstrated the predominance of non-additive genetic effects governing the inheritance of carotenoids [25], other studies reported that carotenoids were additively inherited [11,23,26,27,28,29]. Both additive and non-additive genetic effects also played significant roles in the expression of carotenoids [30,31].

Breeding approaches can improve carotenoids without adverse effects on yield [23,27,28,30]. Unlike sweet and field corn, where multiple studies have investigated the combining ability and heterosis on given parameters, waxy corn lacks similar studies targeting yields and carotenoids. We aim to estimate the combining ability and heterosis of sweet–waxy corn hybrids on yield-related traits and carotenoids. This study will provide insights into heterosis-based biofortification breeding for sweet–waxy corn hybrids with better yield and nutritional values.

## 2. Results

### 2.1. Performance of Parents, F_1_ Hybrids, and Commercial Checks on Yield-Related Traits and Carotenoids

The hybrids exhibited higher means than their corresponding parents for six yield and agronomic traits, except for water-soluble solids and harvest date (Figure 1). The distribution of all carotenoids measured in both hybrids and parents was wide, except for β-cryptoxanthin and β-carotene/β-cryptoxanthin in male lines. Furthermore, there were significant differences between hybrids and their corresponding parents for lutein, β-carotene, β-carotene/β-cryptoxanthin, and β-cryptoxanthin/zeaxanthin. In contrast, the mean values of the hybrids were lower than the parent for total carotenoid content, zeaxanthin, β-cryptoxanthin, and β-carotene/(β-cryptoxanthin + zeaxanthin). There was no difference between hybrids and parents for α-xanthophyll and β-cryptoxanthin + zeaxanthin. These results implied that the different hybrids may exhibit varied performances according to the traits observed.

Parents, hybrids, and commercial checks showed significant differences in all the traits observed (Appendix A). However, we did not notice superior hybrids with high means of yield-related traits and carotenoids. The hybrid ILS_2_ × ILW_1_ had the highest husked ear yield (25.37 ton/ha), surpassing both commercial checks 1 and 2 (1.15 and 1.42-fold, respectively) (Appendix A). The hybrid ILS_4_ × ILW_1_ (20.75 cm) showed the highest husked ear length but was not significantly different from commercial check 1 (19.74 cm). Commercial check 1, a super sweet corn hybrid, had the highest water-soluble solid (13.36 °Brix) but was not significantly different with ILS_5_ × ILW_1_ (12.25 °Brix). The commercial check 1 was the tallest (212.17 cm), while the hybrid ILS_8_ × ILW_2_ was the shortest (163.67 cm). The hybrid ILS_8_ × ILW_2_ exhibited the earliest maturity among all tested hybrids (56.00 days after pollination; DAP), although it was still later than the commercial check 2 (55.17 DAP).

The best hybrid evaluated, ILS_2_ × ILW_1_, had a significantly higher total carotenoid content (7.57 µg/g of FW) than commercial checks 1 and 2 (2.07 and 18.46-fold, respectively) (Appendix A). This hybrid also showed the highest zeaxanthin, β-carotene, α-xanthophyll, and β-cryptoxanthin + zeaxanthin content, surpassing commercial check 1 by 1.71 and 2.33-fold, respectively. The other hybrid, ILS_6_ × ILW_2_, showed higher lutein (3.79 µg/g of FW) content than commercial check 1 by 4.62-fold. The hybrid ILS_3_ × ILW_2_ showed the highest β-cryptoxanthin, β-cryptoxanthin/zeaxanthin, and β-carotene/(β-cryptoxanthin + zeaxanthin) content among all tested hybrids and surpassed commercial check 1 by 1.43–2.21 fold. The hybrid ILS_3_ × ILW_1_ had a higher β-carotene/β-cryptoxanthin content than commercial check 1 by 4.15-fold. Two of the top five hybrids, ILS_2_ × ILW_1_ and ILS_3_ × ILW_2_, exhibited high contents of all observed carotenoids, excluding lutein and β-carotene/β-cryptoxanthin, making these hybrids promising for providing biofortified sweet–waxy corn cultivars.

### 2.2. Variance Components and Heritability Estimates on Yield-Related Traits and Carotenoids

Environment (E), hybrid (H), and their interaction (H × E) were highly significant for yield-related traits and carotenoids, except for husked ear diameter, which was not significant for the H × E (Table 1 and Table 2). We found remarkable variations of GCA_males_, GCA_females_, SCA, and H × E for all studied traits.

The proportion of additive variance to the total variance was predominant for yield-related traits and carotenoids, except for husked ear yield, harvest date, β-cryptoxanthin, and β-cryptoxanthin/zeaxanthin (Table 1 and Table 2). The results indicated that the additive genetic effect was vital in controlling those traits. We found diverse estimates of narrow-sense heritability (*h*^2^*_ns_*), ranging from 0.01 to 0.77, for all studied traits. While husked ear yield, harvest date, β-cryptoxanthin, and β-cryptoxanthin/zeaxanthin had low *h*^2^*_ns_*, the other yield-related traits and carotenoids showed relatively moderate to high *h*^2^*_ns_*.

### 2.3. General Combining Ability (GCA) Effects on Yield-Related Traits and Carotenoids

The GCA estimates varied across parental lines within the same trait and across different traits within the same line (Table 3 and Table 4). We did not obtain any individual line with favorable GCA estimates on all traits studied. Negative GCA effects were desirable for plant height and harvest date, while positive GCA effects were commonly preferred for yield components and carotenoids. Two lines, ILS_8_ and ILW_2_, exhibited significantly negative GCA effects for both plant height and harvest date, making them potential for reducing plant size and shortening maturity (Table 3). The female line ILS_2_ exhibited positive and significant GCA effects for husked ear yield and water-soluble solids. Similarly, the other female line, ILS_7_, had significant GCA effects for husked ear yield and diameter. Those two lines were promising females for increasing yield- and quality-related traits. Meanwhile, the male line ILW_1_ had positive and significant GCA effects on husked ear yield and the other yield components. We can employ that line to develop high-yielding hybrids via heterotic breeding. However, no male lines had positive and significant GCA effects on water-soluble solids.

Two female lines, ILS_3_ and ILS_5_, had positive and significant GCA effects for lutein, β-cryptoxanthin, β-cryptoxanthin/zeaxanthin, and β-carotene/(β-cryptoxanthin + zeaxanthin), whereas the ILS_7_ for total carotenoid content, zeaxanthin, β-carotene, and β-cryptoxanthin + zeaxanthin (Table 4). This finding showed that these females were promising for high carotenoid contents. The male line, ILW_1_, had positive and significant GCA effects for total carotenoid content, zeaxanthin, β-carotene, α-xanthophyll, β-cryptoxanthin + zeaxanthin, and β-carotene/β-cryptoxanthin. In contrast, the other male line, ILW_2_, had favorable GCA for lutein, β-cryptoxanthin, β-cryptoxanthin/zeaxanthin, and β-carotene/(β-cryptoxanthin + zeaxanthin). Considering the GCA effects of different traits and choosing parents that show superiority for the desired heterotic traits, following breeding program objectives are necessary.

### 2.4. Specific Combining Ability (SCA) and Heterosis Effect on Yield-Related Traits and Carotenoids

The SCA effect of hybrids was significant on all observed traits; however, the distribution of the SCA was narrow, except for husked ear yield and plant height (Figure 2a). None of the individual hybrids showed favorable SCA for all traits observed (Appendix A). Negative SCA was important for plant height and harvest date, whereas positive SCA was preferred for yield components and carotenoids. Seven hybrids, including ILS_1_ × ILW_2_, ILS_2_ × ILW_2_, ILS_3_ × ILW_3_, ILS_4_ × ILW_1_, ILS_4_ × ILW_3_, ILS_5_ × ILW_3_, and ILS_6_ × ILW_1_, displayed negative and significant SCA effects for plant height and harvest date representing the short and early maturing hybrids. Four hybrids, including ILS_1_ × ILW_1_, ILS_2_ × ILW_1_, ILS_6_ × ILW_2_, and ILS_6_ × ILW_3_, exhibited positive and significant SCA effects for husked ear yield, husked ear diameter, and husked ear length representing the high yielding hybrids. Two hybrids, ILS_2_ × ILW_3_, and ILS_6_ × ILW_1_, had negative and significant SCA effects for those traits. In addition, ILS_6_ × ILW_1_ displayed negative and significant SCA effects for all traits studied.

Some hybrids showed remarkable SCA effects for carotenoids (Appendix A). Eight hybrids, including ILS_1_ × ILW_1_, ILS_2_ × ILW_1_, ILS_4_ × ILW_2_, ILS_5_ × ILW_2_, ILS_5_ × ILW_3_, ILS_6_ × ILW_2_, ILS_6_ × ILW_3_, and ILS_8_ × ILW_1_, exhibited favorable SCA effects for six of ten carotenoids’ attributes. The hybrid ILS_8_ × ILW_1_ had the highest SCA effects on total carotenoid content, zeaxanthin, α-xanthophyll, and β-cryptoxanthin + zeaxanthin. The hybrid ILS_3_ × ILW_1_ exhibited the highest SCA effects for lutein and β-carotene/β-Cryptoxanthin. The hybrid ILS_3_ × ILW_2_ displayed the highest SCA effects for β-cryptoxanthin and β-carotene/(β-cryptoxanthin + zeaxanthin). Two hybrids, ILS_2_ × ILW_1_, and ILS_3_ × ILW_1_ exhibited the highest SCA effects for β-carotene and β-carotene/β-cryptoxanthin, respectively. Six hybrids, including ILS_1_ × ILW_2_, ILS_2_ × ILW_2_, ILS_3_ × ILW_3_, ILS_5_ × ILW_1_, ILS_6_ × ILW_1_, ILS_7_ × ILW_2_, and ILS_8_ × ILW_3_, exhibited negative and significant SCA effects for 6 of 10 carotenoid traits.

Significant heterosis was observed among hybrids for all traits studied (Figure 2b,c). The husked ear yield, husked ear diameter, husked ear length, and plant height traits revealed positive mid-parent heterosis (mpH). The distribution of mpH for husked ear yield, lutein, β-cryptoxanthin + zeaxanthin, β-cryptoxanthin/zeaxanthin, and β-carotene/(β-cryptoxanthin + zeaxanthin) was broad (Figure 2b). Positive better-parent heterosis (bpH) was found on husked ear yield and husked ear length. The values distributed widely for husked ear yield, β-cryptoxanthin + zeaxanthin, and β-carotene/(β-cryptoxanthin + zeaxanthin) (Figure 2c).

All hybrids demonstrated negative and significant heterosis for water-soluble solid and harvest date, except ILS_1_ × ILW_2_, ILS_5_ × ILW_1_, and ILS_7_ × ILW_3_ hybrids, which had no significance for mpH (Appendix A). Furthermore, all hybrids exhibited positive and significant heterosis for plant height, except ILS_4_ × ILW_2_ and ILS_4_ × ILW_3_ hybrids, which had no significance. The ILS_8_ × ILW_2_, ILS_8_ × ILW_3_, and ILS_1_ × ILW_1_ had high heterosis for husked ear yield, husked ear diameter, husked ear length, and water-soluble solids. The hybrid ILS_2_ × ILW_1_ showed the highest heterosis for total carotenoid content, zeaxanthin, β-carotene, α-xanthophyll, and β-cryptoxanthin + zeaxanthin (Appendix A). The highest heterosis for lutein, β-cryptoxanthin, β-carotene/β-cryptoxanthin, β-cryptoxanthin/zeaxanthin, and β-carotene/(β-cryptoxanthin + zeaxanthin) were found in the hybrids ILS_6_ × ILW_2_, ILS_3_ × ILW_2_, ILS_3_ × ILW_1_, ILS_8_ × ILW_2_, and ILS_5_ × ILW_2_, respectively. One of the 24 hybrids, ILS_3_ × ILW_2_, displayed positive and significant heterosis for β-cryptoxanthin.

### 2.5. Correlation between Yield-Related Traits of F_1_ Hybrids, Heterosis, and Combining Ability

Mid-parent (mpH) and better-parent (bpH) heterosis significantly correlated with the sum of parental general combining ability (GCA_sum_) for harvest date and all carotenoid fractions, except for β-carotene and β-cryptoxanthin/zeaxanthin (Table 5). Moreover, mpH and bpH significantly correlated with SCA for water-soluble solids, plant height, harvest date, total carotenoid content, β-cryptoxanthin, α-xanthophyll, and β-carotene/β-cryptoxanthin. The result implies that the GCA_sum_ is more accurate for predicting heterosis than SCA. The correlation between F_1_ performance and GCA_sum_ was significant and positive for all traits observed. The correlation between F_1_ performance and SCA was also significant and positive for most traits studied, except for husked ear diameter, plant height, lutein, zeaxanthin, β-cryptoxanthin + zeaxanthin, and β-cryptoxanthin/zeaxanthin. Likewise, the correlation between F_1_ performance and heterosis was significant, except for the correlation between F_1_ performance and bpH, which was not significant for most yield-related traits. We found that neither mpH nor bpH were significantly correlated with GCA_sum_ and SCA. Additionally, there was no correlation between F_1_ performance and bpH for husked ear yield, husked ear diameter, or husked ear length.

## 3. Discussion

Lutein and zeaxanthin, which are α-xanthophyll or macular carotenoids central to reducing the risk of AMD, were the predominant carotenoids found in our biofortified hybrids (37.8 and 35.3%, respectively, totaling 73.1%), followed by β-carotene (17.1%) and β-cryptoxanthin (8.7%) (Figure 1 and Appendix A). Previous studies reported that about 30% of lutein or zeaxanthin was found in the F_1_ hybrids evaluated [11,25]. However, other studies found only zeaxanthin as the major carotenoid, representing more than 50% of total carotenoids [15,18,23,27]. Regarding carotenoids central to alleviating vitamin A deficiency, β-carotene was more predominant than β-cryptoxanthin in composing provitamin A. Moreover, 21 of 24 hybrids also showed a higher ratio of β-carotene/β-cryptoxanthin than 1. Our study agreed with the results of Azmach et al. [23], Senete et al. [28], and Owens et al. [32] but was opposed to other studies that reported a larger proportion of β-cryptoxanthin than β-carotene [11,17,27]. These differences may have resulted from the selection for carotenoids, which was carried out during inbred line improvement, or may be due to general differences in the genetic background of germplasm. Differences in extraction and analysis methods may also contribute to differences in carotenoid profiles in field corn [11]. The additional derivative traits, such as the sum and the ratio between individual fractions of carotenoids, may serve as an indirect selection for final carotenoids in corn [32,33]. The following ratios, β-carotene/β-cryptoxanthin, β-cryptoxanthin/zeaxanthin, and β-carotene/(β-cryptoxanthin + zeaxanthin), shared the same β-arm of the biosynthetic pathway; thus, it should be feasible to increase the levels of multiple carotenoids simultaneously. Our study implies that breeding for biofortified corn can include parents expressing substantial and multiple carotenoid compositions.

People recognize traditional waxy corn for its high stickiness due to its high proportion of amylopectin. Today’s consumers prefer more palatable corn with balanced flavor, texture, and aroma [34]. Corn breeders in Thailand utilize the *sh2* recessive genes encoding sweetness to improve the eating quality of traditional waxy corn via sweet–waxy corn hybrids [4,5,6,35]. The biofortified orange waxy corn offers more beneficial values, as carotenoids are vital in maintaining human health. We, therefore, improved high-yielding synergistic waxy corn hybrids carrying double-recessive genes coupled with high carotenoid content. The ILS_2_ × ILW_1_ hybrid was the most promising hybrid among others in this study because it exhibited the highest husked ear yield of 25.37 ton/ha, surpassing both commercial checks. Moreover, it had shorter plant height and earlier maturity than the sweet corn check (Appendix A). This ideotype was suitable for modern corn farming to decrease the percentage of lodging, enable high planting density, improve light interception of the plant canopy, and obtain higher economic yield [34]. The rapid adoption of that hybrid may help corn growers minimize the risk of yield losses due to plant lodging during vegetative and grain-filling stages [34,36].

Water-soluble solids indirectly represent sweetness in vegetable corn. The ILS_2_ × ILW_1_ hybrid also had a higher water-soluble solid than the sweet–waxy corn check but could not surpass the sweet corn check. Consumers who consume steamed waxy corn in their diets prefer the improved sweet–waxy corn hybrid with a strong sweet flavor while maintaining stickiness. In addition to having substantial water-soluble solid, that hybrid had the highest total carotenoid content of 7.57 µg/g of FW (111 µg/g of DW, considering 75% moisture content), which comprised ca. 61.69% zeaxanthin, higher than yellow sweet corn checks at ca. 2.57-fold (Appendix A). Our hybrid also had higher total carotenoid content and zeaxanthin than central Croatian commercial sweet corn hybrids at ca. 4.44 and 1.85-fold, respectively [37]. However, our hybrid could not beat the improved zeaxanthin sweet corn that had a higher value at 2.01-fold [15,18]. Carotenoid content depends on genotype, site-specific pedo-climatic conditions, agronomic factors, nitrogen fertilization [38], and extraction and analysis methods [11]. Our hybrid also revealed the highest β-carotene, α-xanthophyll, and β-cryptoxanthin + zeaxanthin, surpassing the sweet corn hybrid check, accounting from 2.31 to 2.57-fold. The other two hybrids, ILS_6_ × ILW_2_ and ILS_3_ × ILW_2_ were also favorable due to high lutein and β-cryptoxanthin, respectively. Those hybrids mentioned above require further field evaluations over multiple locations and years to confirm their adaptability and stability.

Selecting superior parents enhances the possibility of developing biofortified sweet–waxy corn hybrids. A thorough study of combining ability was essential for understanding genetic effects responsible for yield-related traits and carotenoids. Our study revealed that the additive gene action had remarkable effects in expressing yield-related traits and carotenoids. In contrast, the non-additive gene action predominantly affected the expression of husked ear yield, harvest date, β-cryptoxanthin, and the β-cryptoxanthin/zeaxanthin ratio (Table 1 and Table 2). Dermail et al. [22] found equal contributions between additive and non-additive genetic effects regulating yield-related traits in field corn. Previous investigations reported the immense contribution of additive gene effect instead of non-additive effects on lutein, zeaxanthin, β-cryptoxanthin, and β-carotene of maize [27,29]. Halilu et al. [30] found the predominance of additive gene action on β-cryptoxanthin, whereas non-additive gene actions on grain yield, α-carotene, β-carotene, and provitamin A [11,39]. Meanwhile, both additive and non-additive gene actions controlled carotenoids and their related compounds in the kernels of field corn [40]. Genotype-dependent and environmental effects may explain those contrasting results. Babu et al. [41] noticed that partial-dominant and -recessive gene actions were predominant in corn kernels for the genes *LCYE-50TE* and *crtrB1-30TE*, respectively. The superiority of additive and non-additive gene actions implies applying recurrent selection and heterosis breeding, simultaneously improving targeted traits in corn.

Genetic improvement in crop plants depends on the magnitude of heritability of economic traits [42]. High heritability indicates that the influence of genetic factors is more significant for phenotypes when compared to the environment. Moderate to high heritability estimates were reported for waxy corn yield-related traits [43]. Our present study found narrow-sense heritability ranging from 0.01 to 0.77 for yield and its associated traits (Table 1). Those values were relatively high, indicating the significant progress of breeding for the formation of corn hybrids with suitable ear components and plant height, except for husked ear yield and harvest date. The lack of additive gene effect and poor heritability on husked ear yield and harvest date indicated that slow progress in genetic gain and phenotypic selection could have improved yield and harvest date more effectively. Furthermore, we also noticed that most carotenoids, except β-cryptoxanthin and β-cryptoxanthin/zeaxanthin, illustrated moderate to high narrow-sense heritability. The result corroborated previous investigations on carotenoids in field corn [11,29,40,44,45]. High heritability estimates indicate a higher frequency of favorable alleles and genes controlling the traits and the potential to improve these traits with traditional breeding strategies [46]. Accordingly, heritability observed for carotenoids indicated that conventional breeding is doable for enhancing these traits. However, other studies found that low estimates of heritability were noticed on carotenoids [30,47,48]. These results confirmed that this genetic parameter could be varied for different genetic materials and growing environments. Moreover, the relatively lower heritability of β-cryptoxanthin may be due to technical limitations in reliably separating them from other carotenoids that overlap in the elution system of HPLC [32].

Combining ability helps better understand the mode of gene action controlling the trait of interest and devise breeding strategies to improve the traits. Both the parental lines and their hybrids showed broad ranges of variation. In most, a parent is regarded as a good general combiner if it has higher positive or negative substantial general combining ability (GCA) effects depending on the breeding objectives [49]. Inbreds with significant GCA effects for more than one trait are of great interest for breeding. The female ILS_2_ and ILS_7_ presented positive GCA effects for yield-related traits, and the male was ILW_1_ (Table 3). This result indicated that these inbreds were good in general for yield and their attributes and can be used to develop high-yielding hybrids by sharing desirable alleles. Contrary to females, no males had positive GCA effects for water-soluble solids. It implied that these parents corresponded to the sweet corn and waxy corn groups, respectively, according to Fuengtee et al. [35]. Furthermore, the positive and significant GCA effects for each fraction of carotenoids were separately found in the ILS_3_, ILS_5_, and ILS_7_ females and ILW_1_ and ILW_2_ males, indicating that none of the parental lines were the best general combiner for all the studied traits (Table 4). Meanwhile, the genotypes ILS8 and ILW2 had negative GCA effects for plant height and harvest date, indicating that these lines were potential genetic stocks for short plant stature and early maturity in corn hybrid breeding. Specific combining ability (SCA) effects help identify specific crosses with desirable traits [50]. In this study, the ILS_1_ × ILW_1_ (high × high combiner) hybrid on husked ear yield, zeaxanthin, and β-cryptoxanthin + zeaxanthin had the highest positive SCA effects, caused by additive × additive gene action (Appendix A). The ILS_6_ × ILW_2_ (high × low combiner) hybrid on husked ear diameter, ILS_2_ × ILW_1_ (low × high combiner) hybrid on husked ear length, ILS_8_ × ILW_1_ (low × high combiner) hybrid on total carotenoid content, zeaxanthin, and α-xanthophyll, ILS_6_ × ILW_2_ (low × high combiner) hybrid on lutein, ILS_5_ × ILW_3_ (high × low combiner) hybrid on β-cryptoxanthin, and ILS_2_ × ILW_1_ (high × low combiner) hybrid on β-cryptoxanthin/zeaxanthin, had the highest positive SCA effects due to the epistatic × additive or additive × epistatic mode of gene action. However, the low × low combiners, including ILS_6_ × ILW_3_ and ILS_8_ × ILW_2_ hybrids on water-soluble solid, ILS_2_ × ILW_1_ hybrid on β-carotene, ILS_5_ × ILW_2_ hybrid on β-carotene/β-cryptoxanthin, ILS_5_ × ILW_1_ hybrid on β-cryptoxanthin/zeaxanthin, and ILS_4_ × ILW_1_ hybrid on β-carotene/(β-cryptoxanthin + zeaxanthin), had the highest positive SCA effects due to the presence of dominant × dominant gene action. The development of superior hybrids required any combinations with favorable SCA effects. Parents with high × high, high × low, and low × low GCA effects on traits suggest the presence of additive, dominant, and epistatic gene effects, respectively. The genetic variation in the parents, as measured by the number of heterozygous loci of the parents resembled in the hybrid, may be responsible for the superior hybrids using high × low or low × low GCA effects as parents [49]. For instance, the negative SCA effect desired for the hybrid with short plant stature and earliness could be improved by using transgressive segregants from crosses involving low × low or high combinations of parents. A few hybrids exhibited unfavorable SCA effects on some traits, which might be attributed to the insufficient complementation of parental genes with favorable GCA effects. In contrast, parents with poor GCA effects may produce hybrids with high SCA effects due to the involvement of complementary genes. Previous studies found similar findings [25,27,51]. The high × low combiner was appropriate for heterosis breeding, whereas the high × high combiner for population improvements via pedigree selection [23].

The estimation of the magnitude of heterosis allowed us to identify different cross-combinations, improving the performance of the traits under study. Although there are still some gaps in our understanding of the mechanism of heterosis, significant progress has been made in predicting hybrid performance [52,53]. The ILS_8_ × ILW_2_ hybrid exhibited significantly higher husked ear yield than the corresponding mid- and better-parents. In contrast, the ILS_1_ × ILW_1_ hybrid produced a greater husked ear diameter and length than the corresponding mid- and better parents. Most hybrids had lower means of water-soluble solid and harvest date than their corresponding parents (Appendix A). For carotenoids, none outperformed for all traits studied (Appendix A). Although there was a possibility of exploiting heterosis to increase the concentration of carotenoids [23,27,28,44], some studies reported that heterosis was rare for carotenoids, and this phenomenon could be explained by the QTL approach [54]. The ILS_2_ × ILW_1_ hybrid revealed higher contents of zeaxanthin, β-carotene, α-xanthophyll, and β-cryptoxanthin + zeaxanthin than the corresponding mid- and better-parent. The other hybrids, including ILS_6_ × ILW_2_, ILS_3_ × ILW_2_, ILS_3_ × ILW_1_, ILS_8_ × ILW_2_, and ILS_5_ × ILW_2_, had significantly high heterosis for lutein, β-cryptoxanthin, β-carotene/β-cryptoxanthin, β-cryptoxanthin/zeaxanthin, and β-carotene/(β-cryptoxanthin + zeaxanthin). Thus, we can further explore those hybrids for greater yield, agronomic traits, and nutritional values. Among those hybrids, ILS_2_ × ILW_1_ was the most superior for yield-related traits and carotenoid contents; moreover, this hybrid displayed significantly high estimates of both SCA and heterosis.

Integrating combining ability, hybrid performance, and heterosis helps identify crosses with comparatively high levels of heterosis and thus provides valuable insights for crop improvement. For all traits studied, the relationship between GCA_sum_ and hybrid performance was generally more substantial than between the SCA effect and hybrid performance (Table 5). Therefore, the GCA_sum_ values may be a good indicator for predicting hybrid performance to develop potential hybrids in commercial corn breeding, supported by several previous studies [55]. Moreover, the correlation of GCA_sum_ with the hybrid performance was higher than that with heterosis. We also found that hybrid performance had a stronger correlation with heterosis because heterosis predominantly contributes to trait performance in F_1_ hybrids [56,57]. In contrast, the correlation between the SCA effect and heterosis for most traits studied was insignificant. Hence, the SCA effect may not necessarily be a reliable indicator of heterosis prediction. Other studies reported that dominance effects and nonallelic interactions mainly cause heterosis; therefore, SCA is essential for heterosis [58,59]. Parental adaptation also played an essential role in explaining the high heterosis estimates when the observed traits lacked non-additive gene effects [53].

## 4. Materials and Methods

### 4.1. Plant Materials and Mating Design

Seven of eight sweet corn lines used as females derived from the founder parent genotype Hibrix-53//KV/Delectable carrying double recessive genes (*sh_2_sh_2_wxwx*). This inbred line was developed from tropical waxy corn KV (*Sh_2_Sh_2_wxwx*) and temperate super sweet corn Delectable (*sh_2_sh_2_WxWx*). The progenies were crossed and then backcrossed to tropical sweet corn Hibrix-53 (*sh_2_sh_2_WxWx*) to improve agronomic adaptation and plant stand under tropical climate. Both conventional and SSR marker-assisted selections were performed during family improvements [4,60] at the Thammasat University, Thailand, from 2016 to 2021. Two genotypes, 22-7 (*sh_2_sh_2_wxwx*) and 301-6 (*Sh_2_Sh_2_wxwx*), were obtained from the local seed company. The other two genotypes, 13A-5 and KV3473-2-2 (*Sh_2_Sh_2_wxwx*), differing in kernel colors, were derived from the Plant Breeding Research Center for Sustainable Center, Khon Kaen University, Thailand (Table 6).

To generate synergistic sweet–waxy corn hybrids, eight sweet corn lines were designated as Group I and three waxy corn lines as Group II to generate 24 F_1_ hybrids by following the North Carolina Design II [61]. Those hybrids were established at the Research Farm, Thammasat University, Thailand, in 2021. Due to the different maturity levels of our parental lines, twice to thrice staggered plantings of sweet and waxy corn lines were conducted to ensure pollination [34].

### 4.2. Field Experiment

Eleven parental lines, 24 F_1_ progenies, and two commercial super sweet and sweet–waxy corn hybrids were laid out in a randomized complete block design (RCBD) with three replications and evaluated in the dry season of 2021/22 and the rainy season of 2022 at the Research Farm, Thammasat University, Thailand (+14.07450, +0.6094167, and 7.3 masl). This site had clay soil (pH = 4.91), deficient total nitrogen (0.08%), available phosphorus (3.85 ppm), and high extractable potassium (165.96 ppm). Weather data, including total rainfall, relative humidity, temperature, and solar radiation, were collected from the nearest meteorological stations (Appendix A). Parental lines and hybrids were planted in adjacent blocks in the same field; therefore, these blocks were separately randomized within each replicate. This modification was performed to avoid drawbacks such as borders, shading, and competition effects. Each plot consisted of 4 rows of 5 m in length with 75 cm and 25 cm row and plant spacing, respectively. The crop field management followed the recommendations of the Department of Agriculture, Thailand, including fertilization, irrigation, and pest control. Hand-pollination was carried out to avoid unintended pollen contamination.

### 4.3. Data Collection

The green ears were harvested at approximately 20 to 23 DAP when the corn ears reached the milk stage [62]. The following yield-related traits were observed: plant height, as the average height of ten plants measured from ground level to the base of tassel (m); harvest date, as the number of days from planting to harvesting in 50% of the plants in the plot (days); husked ear yield, as the total weight of the husked ears per plot (ton/ha). Five marketable corn ears were sampled in each plot for measuring the following traits: husked ear diameter, as the average diameter of the five husked ears (cm); husked ear length, as the average length of the five husked ears (cm); and water-soluble solid, as measured using a digital hand-held pocket refractometer (mod. PAL-1, Atago Co., Ltd., Tokyo, Japan) (°Brix).

Five sib-pollinated ears per plot were used as a sample for carotenoid analysis. Kernels located in the middle of cobs were manually separated, frozen in liquid nitrogen to stop the enzymatic activity, and then ground in a sample mill, thoroughly mixed, and stored at −20 °C until analysis.

### 4.4. Sample Preparation and Carotenoid Analysis

The sample extraction followed the Schaub et al. [63] method with slight modifications. The milled samples were transferred to 6 mL of ethanol (containing 0.1% BHT) and mixed with a vortex mixer. Samples were heated in hot water at 85 °C for 3 min and then shaken, and this step was repeated twice. Samples were saponified with 120 µL of 80% KOH and shaken gently by hand. Samples were placed in an ice bath for 5 min, and then added with 4 mL of DI water, followed by thorough mixing using the vortex mixer. Samples were added with 3 mL of diethyl ether (DE)/petroleum ether (PE) (1:1, *v*/*v*) and carefully shaken until the two layers separated. Then, the aqueous solution was transferred into a new test tube. This step was repeated twice, and the resulting layers were pooled. The solution was adjusted to the final volume of 10 mL with PE:DE. The extracted solution was divided equally into two factions for different purposes. The first fraction was used to determine the total carotenoid content of each sample. A UV-vis spectrophotometer (mod. UV-128, Shimadzu Co., Ltd., Tokyo, Japan) was used to measure the absorbance at 450 nm. The total carotenoid content was expressed as micrograms per gram of fresh weight (µg/g of FW). Total carotenoid content was calculated using the following formula:Total carotenoid content (µg/g of FW) = (A × V × 104)/(A^1%^ × g)(1)
where A = absorbance at 450 nm, V = total volume of extract, g = sample weight, and A^1%^ = the extinction coefficient for a mixture of solvents arbitrarily set at 2500.

The second fraction was used to quantify each carotenoid. The extracts were concentrated until dryness under nitrogen flux. Afterward, samples were stored at −20 °C until further analysis. The frozen carotenoid extract was redissolved in 1 mL of methyl tert-butyl ether (MTBE): methanol (75: 25, *v*/*v*) and filtered through a 0.45 µm nylon membrane filter. The composition of solvents and the gradient elution conditions used were described by Wasuwatnakul et al. [25] and Gupta et al. [64] with modifications. Reversed-phase HPLC analysis of carotenoids was performed using a Shimadzu system (Shimadzu Co., Ltd., Tokyo, Japan) equipped with a binary pump (mod. LC-20AC pump) and a diode array detector (mod. SPD-M20A). The HPLC separation was performed on a reversed-phase C_30_ column (250 × 4.6 mm, Ø 3 µm) coupled to a 20 × 4.6 mm C_30_ guard column (YMC Co., Ltd., Kyoto, Japan). Operating conditions were as follows: flow rate of 1.5 mL/min, column temperature of 25 °C, injection volume of 20 µL, and a detection wavelength of 350–600 nm. The mobile phases used were methanol (phase A) and MTBE (phase B). Gradient elution was 50% B at 0 min, followed by a linear gradient to 60% B to 7.00 min at a flow rate of 1.5 mL/min. The 12.10 min gradient was changed to 15% B and was returned to the initial condition by 16.00 min. The four carotenoids, including lutein, zeaxanthin, β-carotene, and β-cryptoxanthin, were identified based on the same retention time and absorption spectral characteristics of external standards. The results for the carotenoids were expressed as µg/g of FW. A series of five sums and ratios, including α-xanthophyll (sum of lutein and zeaxanthin), β-cryptoxanthin + zeaxanthin, β-carotene/β-cryptoxanthin, β-cryptoxanthin/zeaxanthin, and β-carotene/(β-cryptoxanthin + zeaxanthin), followed Baseggio et al. [33].

### 4.5. Data Analysis

The data were subjected to a single analysis of variance to check the homogeneity of residual variances [65]. Since error variances were homogeneous, the data over two seasons were combined following the additive model below.
Y*_ijk_* = *µ* + *a_k_* + *b_j_*_(*k*)_ + *g_i_* + *ag_ik_* + *e_ijk_*(2)
where Y*_ijk_* is the observed value of genotype *i* in replication *j* within environment *k*; *µ* is the population mean; *a_k_* is the environment effect *k* (*k* = 1, 2); *b_j_*_(*k*)_ is the effect of block *j* (*j* = 1, 2, 3) in the environment; *g_i_* is the genotype effect *i* (*i* = 1, 2, 3, …, 44); *ag_ik_* is the effect of interaction between genotype *i* and environment *k*; and *e_ijk_* is the random error associated with observation *ijk* medium. Mean comparisons were performed using the least significant difference (LSD)’s test at 0.05 probability level by Statistix version 10.0 (Analytical Software, Tallahassee, FL, USA).

The North Carolina Design II analysis, combining ability and narrow-sense heritability (*h*^2^*_ns_*) for all studied traits, was estimated using the Analysis of Genetic Designs in R (AGD-R) version 5.0 software [66]. The test for significance of combining ability to the parent and hybrid values used Student’s *t*-test at a 0.05 probability level.

Mid- and better-parent heterosis were calculated using the following formula [67]:mpH = [(F_1_ − mp)/mp] × 100bpH = [(F_1_ − bp)/bp] × 100(3)
where mpH is the mid-parent heterosis, bpH is the better-parent heterosis, F_1_ is the hybrid value, mp is the mid-parent value, and bp is the better-parent value. The test shows the significance of mpH and bpH to the hybrid value using Student’s *t*-test at a 0.05 probability level.

Pearson’s correlation coefficients (*r*) were used to analyze the correlation between F_1_ performance, combining ability, and heterosis and tested at 0.05 and 0.01 probability levels.

## 5. Conclusions

The study found that additive and non-additive genetic effects were substantial for yield-related traits and carotenoids. Moderate to high narrow-sense heritability demonstrated the feasibility of breeding biofortified sweet–waxy corn hybrids with favorable agronomic traits and carotenoids. Despite a few parents with favorable GCA, we suggest breeders include a pairwise parent with high × low combiners in heterosis breeding to improve yield and carotenoids. The hybrid ILS_2_ × ILW_1_ was the most promising, and future extended multi-environment trials were required. The GCA_sum_ of their parents can predict heterosis and per se performances on given traits.

## Figures and Tables

**Figure 1 plants-13-00296-f001:**
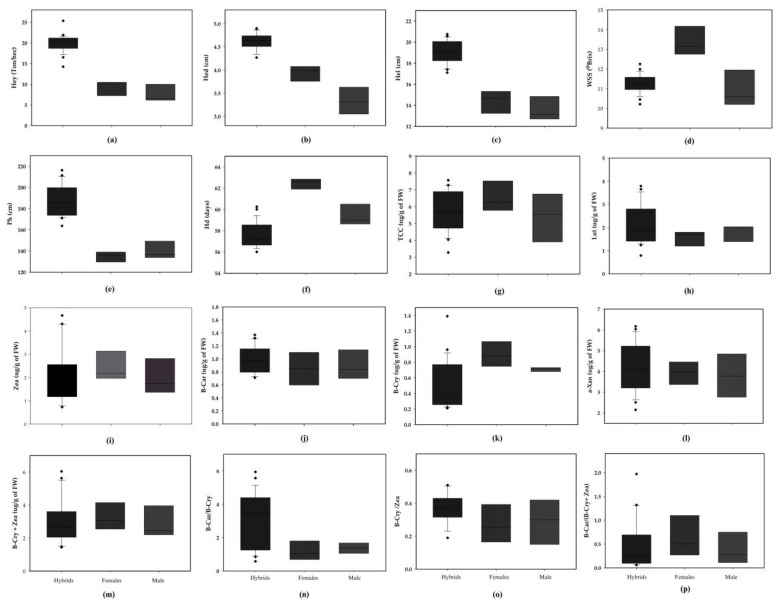
Box plots of the F_1_ hybrids and their corresponding parents on yield-related traits and carotenoids across two seasons between 2021 and 2022. (**a**) Husked ear yield; (**b**) husked ear diameter; (**c**) husked ear length; (**d**) water-soluble solid; (**e**) plant height; (**f**) harvest date; (**g**) total carotenoid content; (**h**) lutein; (**i**) zeaxanthin; (**j**) β-carotene (**k**) β-cryptoxanthin; (**l**) α-xanthophyll; (**m**) β-cryptoxanthin + zeaxanthin; (**n**) β-carotene/β-cryptoxanthin; (**o**) β-cryptoxanthin/zeaxanthin; (**p**) β-carotene/(β-cryptoxanthin + zeaxanthin). The plus sign “

” represents outliers.

**Figure 2 plants-13-00296-f002:**
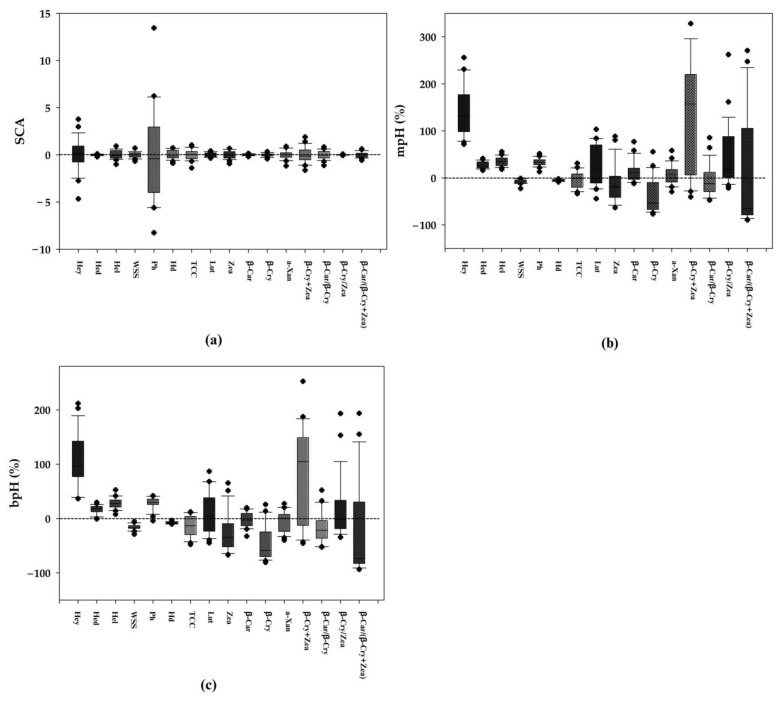
Boxplots of (**a**) specific combining ability (SCA), (**b**) mid-parent heterosis (mpH), and (**c**) better-parent heterosis (bpH) for yield-related traits and carotenoids in 24 hybrids across two seasons between 2021 and 2022. The plus sign “

” represents outliers.

**Table 1 plants-13-00296-t001:** Mean squares for yield-related traits in 24 sweet–waxy corn F_1_ hybrids evaluated across two seasons between 2021 and 2022.

SOV	df	Hey	Hed	Hel	WSS	Ph	Hd
Envi. (E)	1	166.01 **	1.84 **	5.12 **	0.37 **	3974 **	5228 **
Hybrid	23	26.65 **	0.19 **	4.54 **	1.33 **	1256 **	0.56 **
Hybrid × E	23	15.55 **	0.02	1.50 **	0.95 **	125 **	0.19 **
GCA_male_	2	30.35 **	1.49 **	17.51 **	0.72 *	10,256 **	2.39 **
GCA_female_	7	21.23 **	0.13 **	12.43 **	2.29 **	727 **	0.55 **
SCA	14	29.38 **	0.31 **	2.40 **	0.93 **	234 **	0.31 **
GCA_male_ × E	2	23.98 **	0.04 *	1.80 *	6.02 **	301 **	0.19 **
GCA_female_ × E	7	11.71 **	0.03 *	0.98 *	0.42 *	188 **	0.39 **
SCA × E	14	16.26 **	0.19 **	1.71 **	0.49 **	104 **	0.19 **
Pooled error	92	0.39	9.95 × 10^−3^	0.40	0.12	44	1.15 × 10^−3^
σ^2^_A_		0.01	0.77	0.63	0.72	0.77	0.34
σ^2^_D_		0.99	0.23	0.37	0.23	0.23	0.66
*h* ^2^ * _ns_ *		0.01	0.72	0.52	0.77	0.71	0.26

* and **, significant at the 0.05 and 0.01 probability levels, respectively. σ^2^_A_, additive genetic variance; σ^2^_D_, non-additive genetic variance; *h*^2^*_ns_*, narrow-sense heritability. HEY, husked ear yield; Hed, husked ear diameter; Hel, husked ear length; WSS, water-soluble solid; Ph, plant height; Hd, harvest date.

**Table 2 plants-13-00296-t002:** Mean squares for carotenoids in 24 sweet–waxy corn F_1_ hybrids evaluated across two seasons between 2021 and 2022.

SOV	df	TCC	Lut	Zea	β-Car	β-Cry	α-Xan	β-Cry + Zea	β-Car/β-Cry	β-Cry/Zea	β-Car/(β-Cry + Zea)
Envi. (E)	1	13.13 **	5.12 **	3.60 **	4 × 10^−3^ **	6.63 × 10^−3^ **	17.33 **	3.35 **	5.05 **	0.84 **	0.07 **
Hybrid	23	48.67 **	4.54 **	8.60 **	0.25 **	0.56 **	7.65 **	11.13 **	16.57 **	1.45 **	0.05 **
Hybrid × E	23	2.53 **	0.64 **	1.24 **	0.07 **	0.19 **	1.66 **	1.75 **	4.98 **	0.45 **	9.24 × 10^−3^ **
GCA_male_	2	8.29 **	41.50 **	69.94 **	1.53 **	2.39 **	64.99 **	90.09 **	123.97 **	6.24 **	0.36 **
GCA_female_	7	7.41 **	2.10 **	4.92 **	0.23 **	0.55 **	2.26 **	6.49 **	5.83 **	1.42 **	0.04 **
SCA	14	2.95 **	0.49 **	1.68 **	0.07 **	0.31 **	2.16 **	2.17 **	6.60 **	0.79 **	7.52 × 10^−3^ **
GCA_male_ × E	2	0.42 **	0.67 **	0.07 **	0.05 **	0.19 **	0.53 **	0.19 **	7.83 **	1.56 **	0.01 **
GCA_female_ × E	7	2.29 **	0.84 **	0.96 **	0.03 **	0.39 **	1.60 **	1.27 **	6.15 **	0.47 **	0.01 **
SCA × E	14	2.95 **	0.55 **	1.55 **	0.07 **	0.19 **	1.86 **	2.21 **	0.98 **	0.29 **	0.01 **
Pooled	92	7.55 × 10^−3^	3.62 × 10^−3^	2.51 × 10^−3^	4.66 × 10^−4^	1.15 × 10^−3^	8.07 × 10^−3^	3.78 × 10^−3^	0.03	1.46 × 10^−3^	8.69 × 10^−3^
σ^2^_A_		0.53	0.84	0.72	0.60	0.34	0.61	0.72	0.49	0.34	0.77
σ^2^_D_		0.47	0.16	0.28	0.40	0.66	0.39	0.28	0.51	0.64	0.23
*h* ^2^ * _ns_ *		0.41	0.72	0.62	0.49	0.26	0.50	0.62	0.37	0.27	0.63

**, significant at the 0.01 probability level. σ^2^_A_, additive genetic variance; σ^2^_D_, non-additive genetic variance; *h*^2^*_ns_*, narrow-sense heritability. TCC, total carotenoid content; Lut, lutein; Zea, zeaxanthin; β-Car, β-carotene; β-Cry, β-cryptoxanthin; α-Xan, α-xanthophyll.

**Table 3 plants-13-00296-t003:** General combining ability (GCA) of 11 parental lines for yield-related traits evaluated across two seasons between 2021 and 2022.

Parent	Hey	Hed	Hel	WSS	Ph	Hd
ILS_1_	−0.28	−0.07 *	−0.35	0.19 *	2.93	−0.16
ILS_2_	1.06 **	−0.14 **	0.00	0.56 **	2.37	0.03
ILS_3_	−0.25	0.03	−0.27	−0.48 **	−7.02 **	1.56 **
ILS_4_	1.29 **	−0.02	1.23	0.10	13.23 **	−0.50 *
ILS_5_	0.22	−0.06 *	0.99 **	0.16 *	−3.46	1.42 **
ILS_6_	−1.62 **	0.06 *	−0.33	−0.49 **	−0.90	−0.88 **
ILS_7_	0.91 **	0.11 **	0.15	−0.16	−1.85	−0.25
ILS_8_	−1.34 **	0.08 *	−1.42 **	0.11	−5.29 *	−1.22 **
ILW_1_	0.63 *	0.09 *	0.66 **	0.14	15.55 **	0.51 *
ILW_2_	−0.89 **	−0.20 **	−0.13	−0.09	−13.46 **	−0.53 *
ILW_3_	0.27	0.12 **	−0.53 *	−0.05	−2.09	0.02

* and **, GCA estimates are significantly different from zero at ≥SE and ≥2SE, respectively. HEY, husked ear yield; Hed, husked ear diameter; Hel, husked ear length; WSS, water-soluble solid; Ph, plant height; Hd, harvest date. Any inbred lines labeled with ILS and ILW were assigned as females and males, respectively.

**Table 4 plants-13-00296-t004:** General combining ability (GCA) of 11 parental lines for carotenoids evaluated across two seasons between 2021 and 2022.

Parent	TCC	Lut	Zea	β-Car	β-Cry	α-Xan	β-Cry + Zea	β-Car/β-Cry	β-Cry/Zea	β-Car/(β-Cry + Zea)
ILS_1_	0.22	−0.40 *	0.41 *	0.01	−0.13 *	0.01	0.42	0.35	−0.04 *	−0.21 *
ILS_2_	0.52 *	−0.34 *	0.59 **	0.07 *	−0.12 *	0.25	0.66 **	0.09	−0.05 **	−0.26 **
ILS_3_	−0.05	0.37 *	−0.56 *	−0.03	0.29 **	−0.20	−0.59 *	−0.42	0.06 **	0.48 **
ILS_4_	−0.61 **	0.16	−0.48 *	−0.06 *	−0.03	−0.32	−0.54 *	−0.28	0.05 **	0.13
ILS_5_	−0.70 **	0.38 *	−0.68 **	−0.13 **	0.16 **	−0.30	−0.81 **	−0.90 **	0.05 **	0.27 **
ILS_6_	0.50 *	0.16	−0.06	0.08 *	0.14 *	0.10	0.03	0.40	0.01	0.06
ILS_7_	0.91 *	0.12	0.59 **	0.19 **	−0.15 **	0.71 **	0.78 **	0.93 **	−0.03 *	−0.28 **
ILS_8_	−0.79 **	−0.45 **	0.19	−0.14 **	−0.16 **	−0.26	0.05	−0.17	−0.04 *	−0.19 *
ILW_1_	0.95 **	−0.29 *	1.37 **	0.17 **	−0.16 **	1.08 **	1.54 *	1.47 **	−0.10 **	−0.28 **
ILW_2_	0.10	1.04 **	−0.89 **	−0.19 **	0.25 **	0.15	−1.08 **	−1.71 **	0.06 **	0.41 **
ILW_3_	−1.06 **	−0.75 **	−0.48 *	0.02	−0.09	−1.23 **	−0.46 *	0.24	0.04 *	−0.13

* and **, GCA estimates are significantly different from zero at ≥SE and ≥2SE, respectively. TCC, total carotenoid content; Lut, lutein; Zea, zeaxanthin; β-Car, β-carotene; β-Cry, β-cryptoxanthin; α-Xan, α-xanthophyll. Any inbred lines labeled with ILS and ILW were assigned as females and males, respectively.

**Table 5 plants-13-00296-t005:** Correlation analysis of heterosis, combining ability, and hybrid performance (F_1_) for yield-related traits and carotenoids across two seasons between 2021 and 2022.

Traits	mpH-bpH	mpH-GCA_sum_	mpH-SCA	bpH-GCA_sum_	bpH-SCA	F_1_-GCA_sum_	F_1_-SCA	F_1_-mpH	F_1_-bpH
Hey	0.913 **	−0.167	0.395	−0.162	0.312	0.580 *	0.815 **	0.225	0.160
Hed	0.751 **	0.389	−0.040	0.245	0.016	0.932 **	0.160	0.461 *	0.298
Hel	0.856 **	0.353	0.328	−0.002	0.155	0.900 **	0.437 *	0.459 *	0.065
WSS	0.770 **	0.320	0.520 **	0.310	0.449 *	0.757 **	0.656 **	0.583 *	0.528 *
Ph	0.866 **	0.385	0.413 *	0.004	0.291	0.941 **	0.337	0.501 *	0.102
Hd	0.846 **	0.736 **	0.484 *	0.595 *	0.454 *	0.913 **	0.409 *	0.868 **	0.728 **
TCC	0.937 **	0.597 *	0.506 *	0.693 **	0.384	0.885 **	0.467 *	0.764 **	0.792 **
Lut	0.960 **	0.882 **	0.262	0.908 **	0.299	0.967 **	0.253	0.920 **	0.955 **
Zea	0.941 **	0.876 **	0.377	0.894 **	0.349	0.936 **	0.343	0.953 **	0.959 **
β-Car	0.809 **	0.324	0.380	0.296	0.295	0.935 **	0.417 *	0.449 *	0.400 *
β-Cry	0.992 **	0.810 **	0.552 *	0.781 **	0.542 *	0.817 **	0.569 *	0.984 **	0.955 **
α-Xan	0.935 **	0.657 **	0.560 *	0.674 **	0.427 *	0.910 **	0.412 *	0.848 **	0.792 **
β-Cry+Zea	0.964 **	0.836 **	0.312	0.839 **	0.299	0.940 **	0.344	0.918 **	0.891 **
β-Car/β-Cry	0.954 **	0.699 **	0.500 *	0.754 **	0.545 *	0.870 **	0.492 *	0.855 **	0.925 **
β-Cry/Zea	0.955 **	0.389	0.214	0.336	0.196	0.949 **	0.303	0.454 *	0.395 *
β-Car/(β-Cry + Zea)	0.964 **	0.690 **	0.301	0.552 *	0.271	0.802 **	0.574 *	0.758 **	0.608 **

mpH, mid-parent heterosis; bpH, better-parent heterosis; GCA_sum_, the sum of general combining ability for two parents; SCA, specific combining ability. * and ** are significantly different at 0.05 and 0.01 probability levels, respectively. HEY, husked ear yield; Hed, husked ear diameter; Hel, husked ear length; WSS, water-soluble solid; Ph, plant height; Hd, harvest date. TCC, total carotenoid content; Lut, lutein; Zea, zeaxanthin; β-Car, β-carotene; β-Cry, β-cryptoxanthin; α-Xan, α-xanthophyll.

**Table 6 plants-13-00296-t006:** Parental inbred lines used in the North Carolina Design II scheme.

Inbred Line ^1/^	Pedigree	Genotype	Source of Ancestor	Relative Carotenoid Content (μg/g of FW) ^2/^
ILS_1_	Hibrix-53//KV/Delectable-BC_1_-22-1-4-3-1-1	*sh* _2_ *sh* _2_ *wxwx*	Thai/Vietnam/USA	5.21
ILS_2_	Hibrix-53//KV/Delectable-BC_1_-65-5-1-3-1-1	*sh* _2_ *sh* _2_ *wxwx*	Thai/Vietnam/USA	3.97
ILS_3_	Hibrix-53//KV/Delectable-BC_1_-34-1-3-4-6-1	*sh* _2_ *sh* _2_ *wxwx*	Thai/Vietnam/USA	8.01
ILS_4_	Hibrix-53//KV/Delectable-BC_1_-2-1-1-5-3-1	*sh* _2_ *sh* _2_ *wxwx*	Thai/Vietnam/USA	5.75
ILS_5_	Hibrix-53//KV/Delectable-BC_1_-5-5-3-9-7-1	*sh* _2_ *sh* _2_ *wxwx*	Thai/Vietnam/USA	8.61
ILS_6_	Hibrix-53//KV/Delectable-BC_1_-17-4-1-9-1-1	*sh* _2_ *sh* _2_ *wxwx*	Thai/Vietnam/USA	6.04
ILS_7_	Hibrix-53//KV/Delectable-BC_1_-17-4-2-8-5-1	*sh* _2_ *sh* _2_ *wxwx*	Thai/Vietnam/USA	8.11
ILS_8_	22-7	*sh* _2_ *sh* _2_ *wxwx*	Thai (Sweet × Waxy corn)	4.99
ILW_1_	13A-5	*Sh* _2_ *Sh* _2_ *wxwx*	Thai composite #1-5	8.15
ILW_2_	KV3473-2-2	*Sh* _2_ *Sh* _2_ *wxwx*	Thai/USA	5.02
ILW_3_	301-6	*Sh* _2_ *Sh* _2_ *wxwx*	Thai (Sweet × Waxy corn)	4.10
Check 1	Super sweet corn	*sh* _2_ *sh* _2_ *WxWx*		-
Check 2	Sweet–waxy corn	*Sh* _2_ *sh* _2_ *wxwx*		-

^1/^ Any inbred lines labeled with ILS and ILW were assigned as females and males, respectively. ^2/^ Relative carotenoid content derived from preliminary analyses.

## Data Availability

Data are contained within the article and Appendix A.

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
