# Peer review of "The Combining Ability and Heterosis Analysis of Sweet–Waxy Corn Hybrids for Yield-Related Traits and Carotenoids"

_plants, 2024, doi:10.3390/plants13020296_

Round 1

Reviewer 1 Report

Comments and Suggestions for Authors

Kanyarat Prai-Anan and his (her) colleagues aim to estimates the combining ability and heterosis of sweet-waxy corn hybrids for yield related traits and carotenoids, and develop a candidate biofortified sweet-waxy corn hybrid possessing high yields and carotenoids. The submitted manuscript is written clearly and general interest to the readers. However, I have several comments that should be addressed before publication.

In scientific aspects:

1.       Comparing to Specific combining ability (SCA), why the general combining ability (GCA) showed more positively correlated with high yields and carotenoids content. The authors should give a further discussion in the discussion section.  

Comments on the Quality of English Language

In language aspect:

1.   The space (gap) should be inserted between the 2 and mg (Page2, L53), for example, “2 mg per day as suggested”. 

2.Supplemental Tables 1S and 2S” (Page3, L96) suggested to be replaced by “Supplemental Tables S1 and S2”. This is only the example. Similar texts are also seen in many sections of the manuscript. Please checked the whole manuscript and correct them in the revised manuscript and Supplementary Materials.

3.  The semicolon “;” after“……2021 and 2022;” (Page4, L122) should be replaced by full stop “.”

4.  In the discussion section, the sentences “Our study agreed with……, and but……”(Page10, L285-287) and “……and the male ILW1 (Table 3).” (Page10, L377-378) were confused. Please check grammar and improved the readability of these sentences.

5. The authors must carefully check grammar, punctuation, spelling, and overall style in the whole text.

Author Response

Dear: Reviewer #1,

We gratefully appreciate the reviewers for their advice and thoughtful comments made on our manuscript. We have carefully taken the comments into consideration to prepare our revision, which has resulted in a paper that is clearer and more compelling. Below are our responses to the Reviewer’s comments. The line numbers refer to our revised manuscript.

 COMMENT 1: Comparing to Specific combining ability (SCA), why the general combining ability (GCA) showed more positively correlated with high yields and carotenoids content. The authors should give a further discussion in the discussion section. 

RESPONSE 1: As mentioned earlier, hybrid performance is contingent upon the effects of GCA, which are regulated by additive genetic variance. This implies that GCA holds greater significance than SCA in influencing these traits. Consequently, GCA values serve as a reliable indicator for predicting hybrid performance, facilitating the development of potential hybrids in commercial corn breeding (Lines 443-447).

COMMENT 2: The space (gap) should be inserted between the 2 and mg (Page2, L53), for example, “2 mg per day as suggested”.

RESPONSE 2: The space has been inserted into the sentence (Line 53).

COMMENT 3: .“Supplemental Tables 1S and 2S” (Page3, L96) suggested to be replaced by  “Supplemental Tables S1 and S2”. This is only the example. Similar texts are also seen in many sections of the manuscript. Please checked the whole manuscript and correct them in the revised manuscript and Supplementary Materials.

RESPONSE 3: It has been checked and edited in both the revised manuscript (Lines 96,99,109,204,215,242,247,281,310,322,394,427,429,490) and Supplementary Materials.

COMMENT 4: In the discussion section, the sentences “Our study agreed with……, and but……”(Page10, L285-287) and “……and the male ILW1 (Table 3).” (Page10, L377-378) were confused. Please check grammar and improved the readability of these sentences.

RESPONSE 4: These sentences have been checked and edited (Lines 285-287, 379-380).

COMMENT 5: The authors must carefully check grammar, punctuation, spelling, and overall style in the whole text.

RESPONSE 5: We thank the reviewer for pointing this out. The revised manuscript has been checked and edited by all authors and native speaker.

 Regards,

Reviewer 2 Report

Comments and Suggestions for Authors

Review report –ID plants- 2810853

Combining Ability and Heterosis Analysis of Sweet-Waxy Corn Hybrids for Yield-Related Traits and  Carotenoids

Dear authors,

To improve the manuscript, I have some remarks and specific comments:

1.     Abstract section of the manuscript

The abstract section of the manuscript: is well-organized and well-written.

2.     Introduction- section of the manuscript:

-lines 53- 55 ‘’ Other carotenoids found in biofortified corn are provitamin A, including α-carotene, β-carotene, and β-cryptoxanthin, which can be converted into retinol. Please add some references to confirm this statement.

- lines 58-60 the authors stated that biofortified sweet-waxy corn hybrids with high eating quality and carotenoid contents will expand the market segments and provide benefits to human health. Please explain in more detail why increasing nutrient density in staple food crops through plant breeding remains the preferred choice for sustainable agriculture.

-lines 62-63

The per se evaluation is often ineffective for parental selection on yield-related traits due to polygenic [20]. Please rephrase more concisely and mention the authors of the research not just the number in brackets.

4.     Material and methods section of the manuscript

-is well organized and well-written.

- data analysis included methods in agreement with the aims of this manuscript.

Please explain in more detail why you chose in this research the North Carolina mating design II scheme.

2.     Results section of the manuscript – is well organized regarding the performance of Parents, F1 Hybrids, and Commercial Checks on Yield-related Traits and Carotenoids

The subsections

2.3. General Combining Ability (GCA) Effects on Yield-related Traits and Carotenoids and 2.4. Specific Combining Ability (SCA) Effects and Heterosis on Yield-related Traits and Carotenoids can be organized more concisely.

-lines 262-263 Likewise, the correlation between F1 performance and heterosis was significant, except for F1 performance and bpH for most yield-related traits. Please rephrase this more clearly.

3. Discussion section of the manuscript is well-organized and well-written.

I suggest you mention that the need for this research is that unlike sweet and field corn, where several studies have investigated the combining ability and heterosis on given morphological parameters, waxy corn thus does not have similar studies targeting yield parameters and carotenoid contents.

5. Conclusions section of the manuscript

This section of the manuscript is well-written. However, I suggest you briefly emphasize the novelty of this research.

References list

Please add the suggested references (line 55 of the manuscript) to the references list.

Author Response

Dear: Reviewer #2,

We gratefully appreciate the reviewers for their advice and thoughtful comments made on our manuscript. We have carefully taken the comments into consideration to prepare our revision, which has resulted in a paper that is clearer and more compelling. Below are our responses to the Reviewer’s comments. The line numbers refer to our revised manuscript.

 COMMENT 1: lines 53- 55 ‘’ Other carotenoids found in biofortified corn are provitamin A, including α-carotene, β-carotene, and β-cryptoxanthin, which can be converted into retinol. Please add some references to confirm this statement.

RESPONSE 1: The references were added in this sentence (Lines 53-55).

 COMMENT 2: lines 58-60 the authors stated that biofortified sweet-waxy corn hybrids with high eating quality and carotenoid contents will expand the market segments and provide benefits to human health. Please explain in more detail why increasing nutrient density in staple food crops through plant breeding remains the preferred choice for sustainable agriculture.

RESPONSE 2: For the reasons, increasing nutrient density in staple food crops through plant breeding is a holistic and sustainable approach that addresses nutritional, economic, environmental, and cultural considerations, making it a preferred choice for promoting food security and overall well-being.

COMMENT 3: lines 62-63 The per se evaluation is often ineffective for parental selection on yield-related traits due to polygenic [20]. Please rephrase more concisely and mention the authors of the research not just the number in brackets.

RESPONSE 3: This sentence has been checked and edited (Lines 62-63).

COMMENT 4: Please explain in more detail why you chose in this research the North Carolina mating design II scheme.

RESPONSE 4: This mating design permits the estimation of combining ability and variance components, where each member of a group of parents used as males is mated to each member of another group of parents. Thus, eight sweet corn lines were designated as Group I and three waxy corn lines as Group II to generate 24 F1 sweet-waxy corn hybrids (Lines 471–473).

(Singh, R.K.; Chaudhary, B.D. Biometrical Methods in Quantitative Genetic Analysis. Kalyani Publishers, New Delhi, India. 1985.

Muthoni J. and Shimeli, H. Mating designs commonly used in plant breeding: A review. AJCS 2020, 14, 1855-1869)

COMMENT 5: The subsections

2.3. General Combining Ability (GCA) Effects on Yield-related Traits and Carotenoids and 2.4. Specific Combining Ability (SCA) Effects and Heterosis on Yield-related Traits and Carotenoids can be organized more concisely.

RESPONSE 5: The 2.4. sub-section has been edited (Line 200-201).

COMMENT 6: -lines 262-263 Likewise, the correlation between F1 performance and heterosis was significant, except for F1 performance and bpH for most yield-related traits. Please rephrase this more clearly.

RESPONSE 6: This sentence has been checked and edited (Lines 264-266).

COMMENT 7: I suggest you mention that the need for this research is that unlike sweet and field corn, where several studies have investigated the combining ability and heterosis on given morphological parameters, waxy corn thus does not have similar studies targeting yield parameters and carotenoid contents.

RESPONSE 7: Thank you for the reviewer’s suggestion. This point has been addressed in the manuscript (Lines 36-48).

COMMENT 8: Conclusions section of the manuscript

This section of the manuscript is well-written. However, I suggest you briefly emphasize the novelty of this research.

RESPONSE 8: This study is the first time that reports on combining ability and heterosis of yield-related traits and carotenoids in biofortified sweet-waxy corn hybrids. Moreover, GCAsum can predict heterosis for improving biofortified sweet-waxy corn hybrid enrich in carotenoids. This point has been addressed in the abstract and conclusion sections of manuscript (Lines 23-25, 584-585).

COMMENT 9: References list

Please add the suggested references (line 55 of the manuscript) to the references list.

RESPONSE 9: Please see comment 1st.

 Regards,

Reviewer 3 Report

Comments and Suggestions for Authors

Thank you for given me the chance to read the manuscript “Combining Ability and Heterosis Analysis of Sweet-Waxy Corn Hybrids for Yield-Related Traits and Carotenoids”

Sweet-waxy corn is an important cultivar having good commercial future and human health benefit.The article is very interesting, original, the respected authors obtained important results that are of fundamental and, most importantly, practical significance. The content of the article corresponds to the abstract and title. The article is very well written and will be of interest not only to narrow specialists. Tables and figures complement the text well. I would especially like to note the very good quality of the tables illustrating the article. I would also like to note that the article will be in demand and of interest to a wide range of corn breeders. 

Overall, I really liked the article and I believe that it can be accepted for publication.

Author Response

Dear: Reviewer,

We gratefully appreciate the reviewers for their advice and thoughtful comments made on our manuscript. We have carefully taken the comments into consideration to prepare our revision, which has resulted in a paper that is clearer and more compelling. Below are our responses to the Reviewer’s comments.

  COMMENT 1: Sweet-waxy corn is an important cultivar having good commercial future and human health benefit. The article is very interesting, original, the respected authors obtained important results that are of fundamental and, most importantly, practical significance. The content of the article corresponds to the abstract and title. The article is very well written and will be of interest not only to narrow specialists. Tables and figures complement the text well. I would especially like to note the very good quality of the tables illustrating the article. I would also like to note that the article will be in demand and of interest to a wide range of corn breeders.

Overall, I really liked the article and I believe that it can be accepted for publication.

RESPONSE 1: Thank you for your recommendations. We hope that our work proves useful to corn breeders, particularly those focusing on breeding strategies for biofortified sweet-waxy corn hybrids with high yield and carotenoid content.

Regards,
